# Dynamic blood oxygen indices in mechanically ventilated COVID-19 patients with acute hypoxic respiratory failure: A cohort study

Luke Bracegirdle[1], Alexander Jackson[1,2], Ryan Beecham[1], Maria Burova[1], Elsie Hunter[1], Laura G. Hamilton[1], Darshni Pandya[1], Clare Morden[1], Michael P. W. Grocott[1,2,3]*, Andrew Cumpstey[1,2,3], Ahilanandan Dushianthan[1,2,3], the REACT COVID-19 Investigators[¶]

1 General Intensive Care Unit, University Hospital Southampton NHS Foundation Trust, Southampton, Hampshire, United Kingdom, 2 NIHR Southampton Clinical Research Facility and NIHR Southampton Biomedical Research Centre, University Hospital Southampton / University of Southampton, Southampton, Hampshire, United Kingdom, 3 Integrative Physiology and Critical Illness Group, Clinical and Experimental Sciences Faculty of Medicine, University of Southampton, University Hospital Southampton, Southampton, Hampshire, United Kingdom

¶ REACT COVID-19 Investigators can be found in the acknowledgments
* mike.grocott@soton.ac.uk

**Data Availability Statement:** All oxygen indices files are available from the Figshare database https://doi.org/10.6084/m9.figshare.16628182.v1

## Abstract

### Background

Acute hypoxic respiratory failure (AHRF) is a hallmark of severe COVID-19 pneumonia and often requires supplementary oxygen therapy. Critically ill COVID-19 patients may require invasive mechanical ventilation, which carries significant morbidity and mortality. Understanding of the relationship between dynamic changes in blood oxygen indices and clinical variables is lacking. We evaluated the changes in blood oxygen indices–$PaO_2$, $PaO_2/FiO_2$ ratio, oxygen content ($CaO_2$) and oxygen extraction ratio ($O_2ER$) in COVID-19 patients through the first 30-days of intensive care unit admission and explored relationships with clinical outcomes.

### Methods and findings

We performed a retrospective observational cohort study of all adult COVID-19 patients in a single institution requiring invasive mechanical ventilation between March 2020 and March 2021. We collected baseline characteristics, clinical outcomes and blood oxygen indices. 36,383 blood gas data points were analysed from 184 patients over 30-days. Median participant age was 59.5 (IQR 51.0, 67.0), BMI 30.0 (IQR 25.2, 35.5) and the majority were men (62.5%) of white ethnicity (70.1%). Median duration of mechanical ventilation was 15-days (IQR 8, 25). Hospital survival at 30-days was 72.3%. Non-survivors exhibited significantly lower $PaO_2$ throughout intensive care unit admission: day one to day 30 averaged mean difference -0.52 kPa (95% CI: -0.59 to -0.46, p<0.01). Non-survivors exhibited a significantly lower $PaO_2/FiO_2$ ratio with an increased separation over time: day one to day 30 averaged

**Funding:** The authors received no specific funding for this work.

**Competing interests:** The authors have declared that no competeing interests exist.

mean difference -5.64 (95% CI: -5.85 to -5.43, p<0.01). While all patients had sub-physiological $CaO_2$, non-survivors exhibited significantly higher values. Non-survivors also exhibited significantly lower oxygen extraction ratio with an averaged mean difference of -0.08 (95% CI: -0.09 to -0.07, p<0.01) across day one to day 30.

## Conclusions

As a novel cause of acute hypoxic respiratory failure, COVID-19 offers a unique opportunity to study a homogenous cohort of patients with hypoxaemia. In mechanically ventilated adult COVID-19 patients, blood oxygen indices are abnormal with substantial divergence in $PaO_2/FiO_2$ ratio and oxygen extraction ratio between survivors and non-survivors. Despite having higher $CaO_2$ values, non-survivors appear to extract less oxygen implying impaired oxygen utilisation. Further exploratory studies are warranted to evaluate and improve oxygen extraction which may help to improve outcomes in severe hypoxaemic mechanically ventilated COVID-19 patients.

## Introduction

The novel SARS-CoV-2 viral infection (coronavirus disease (COVID-19)) is currently imposing an unprecedented challenge for the medical community worldwide. A global pandemic was declared by the World Health Organisation (WHO) in January 2020 and continues to cause significant burden from multiple waves of varying lineages accounting for around 4.5 million case fatality to date [1]. The majority of patients develop mild illness without any significant respiratory sequelae [2]. Hypoxic respiratory failure is the hallmark of severe COVID-19 pneumonia and often requires supportive oxygen therapy via various delivery methods [3]. Development of Acute Respiratory Distress syndrome (ARDS) and persistent hypoxaemia necessitating admission to an intensive care unit (ICU) for invasive mechanical ventilation carries substantial mortality in the region of 50% [4].

As a novel cause of acute hypoxic respiratory failure, COVID-19 offers a unique opportunity to study a relatively homogeneous cohort of patients with similar underlying pathology. Compared to other critically unwell patients, this group requires high concentrations of inspired oxygen for prolonged periods and tends not to display the typical features of respiratory distress despite profound hypoxia [5]. Moreover, the presence of acute hypoxic respiratory failure and the degree of hypoxaemia, defined by the ratio of arterial partial pressure of oxygen ($PaO_2$) to the fractional inspired oxygen ($PaO_2/FiO_2$ ratio), are independently associated with increased mortality [6]. Consequently, effective oxygen therapy via mechanical ventilation remains the mainstay of critical care management of patients with severe hypoxic respiratory failure. However, it is unclear if increments in the fractional inspired oxygen improve blood oxygen indices such as total arterial oxygen content ($CaO_2$) and oxygen utilisation or impact the overall clinical outcomes of mechanically ventilated COVID-19 patients with severe hypoxic respiratory failure.

One method of assessing tissue-level oxygen utilisation is to examine the balance between oxygen delivery ($DO_2$) and oxygen uptake ($VO_2$), by calculating the oxygen extraction ratio ($O_2ER$). In health, $O_2ER$ at rest is approximately 25% and therefore is usually 'supply independent'. It may increase in well-trained athletes and may exceed 75% under conditions of exceptional metabolic stress. Recent work examining venous oxygen saturation ($SvO_2$) suggests

oxygen extraction may be compromised in patients with COVID-19 and that such compromise may be associated with reduced survival, although this work examined blood oxygen indices immediately after admission to ICU and not throughout the admission course [7]. As both hypoxaemia and hyperoxemia can be associated with adverse outcomes in critically ill patients [8], it is imperative to assess tissue level oxygen availability and extraction. The aim of this study was to describe trends in blood oxygen indices ($PaO_2/FiO_2$ ratio, $CaO_2$ and $O_2ER$) in patients with COVID-19 throughout the first 30-days of intensive care admission and explore the relationship between these indices and clinical outcomes.

## Methods

Ethical approval was provided as part of the REACT COVID-19 observational study (a longitudinal cohort study to facilitate better understanding and management of SARS-CoV-2 infection from hospital admission to discharge across all levels of care): REC reference 17/NW/0632, SRB reference number; SRB0025 [9]. Due to the retrospective and observational nature of the study and there were no identifiable patient's source data, the need for individual informed patient consent was waived. The data analysed were already routinely collected and electronically stored as part of clinical care. All data were anonymised and handled according to the local institutional and national policies. The study used STROBE guidelines for reporting observational studies [10].

We performed a retrospective observational cohort study in a single centre University Teaching Hospital in the UK. We included all patients admitted to the General Intensive Care Unit, between 1st March 2020 and 31st March 2021 inclusive. Eligible participants were aged 18 years or over, tested positive for COVID-19 by reverse transcriptase-polymerase chain reaction (RT-PCR) nasal and throat specimens, required mechanical ventilation, and had one or more arterial blood gas (ABG) samples performed. As a pragmatic retrospective study without intervention, we evaluated the merit of various oxygen indices under a real-life, generalised intensive care setting which may be applicable to routine practice. We therefore did not exclude any patients based on the presence of comorbidities that may have contributed to their death or those enrolled in other clinical trials.

Suitable patients were identified using admission records by a combination of manual and semi-automated data extraction. We collected baseline patient characteristics (age, gender, comorbidities), Clinical Frailty Scale (CFS) [11] and Charlson Comorbidity Index (CCI) [12] and ICU severity indices, including Acute Physiology and Chronic Health Evaluation–II (APACHE II) [13] and sequential organ failure assessment score (SOFA) [14]. The Intensive Care and National Audit Centre (ICNARC) UK summary data were used for comparison [4]. Additional data were extracted from our institution's electronic patient record (EPR) (MetaVision, iMDSoft, Tel Aviv, Israel). At our institution, blood gas and laboratory results, ventilation parameters and vital signs are recorded automatically or by the bedside nurse and stored within the EPR. These were extracted for the entire duration of ICU stay for all included patients. Data underwent exploratory analysis and data cleaning to remove erroneous values and ensure data quality for the final analyses. The median days of invasive mechanical ventilation were 15 (interquartile range 8, 25). Therefore, to capture the entire intubated duration and initial recovery, results are reported for day one to seven and day one to 30. The primary outcome was hospital mortality at 30 days.

$FiO_2$ was extracted directly from the ventilator to avoid labelling errors. $PaO_2/FiO_2$ ratios were calculated for each arterial blood gas sample. Total arterial oxygen content ($CaO_2$) is the sum of the oxygen bound to haemoglobin and oxygen dissolved in plasma. It is calculated by

[15];

$$CaO_2 = (1.34 \text{ x } [Hb] \text{ x } SaO_2) + (0.023 \text{ x } PaO_2)$$

where 1.34 is Hüfner's constant, Hb is the amount of haemoglobin in grams per decilitre ($gdl^1$), $SaO_2$ the arterial haemoglobin saturation in fraction, 0.023 the solubility coefficient of oxygen at body temperature (i.e., the number of millilitres of oxygen dissolved per 100ml of plasma per kilopascal (ml $O_2$ $100ml^{-1}$ plasma $kPa^{-1}$), and $PaO_2$ the partial pressure of oxygen in arterial blood in kilopascals (kPa). Both point of care (ABG) and lab haemoglobin values were available, with laboratory values used across all calculations.

Venous blood gas (VBG) samples taken from central venous catheters within 30 minutes of an ABG on the same $FiO_2$ were studied for central venous saturations ($ScvO_2$), to ensure a strict temporal relationship been arterial and central venous samples, and used as a surrogate marker for pulmonary artery mixed blood saturation ($SvO_2$). All other venous gas samples were excluded. The oxygen content of mixed venous blood ($CvO_2$) was then calculated:

$$CvO_2 = (1.34x \text{ } [Hb] \text{ x } ScvO_2) + (0.023 \text{ x } PvO_2)$$

Oxygen extraction ratio ($O_2ER$) was then calculated by the following equation:

$$O_2ER = CaO_2 - CvO_2/CaO_2$$

These formulas were selected in order to avoid reliance on cardiac output monitoring. All calculations for $O_2ER$ were also compared to those produced when using ($SaO_2 - SvO_2$) / $SaO_2$, producing similar results.

## Statistical analysis

Statistical analysis and data processing were performed using R (R Core Team, Vienna, Austria) and GraphPad Prism version 9.0.0 for Windows, (GraphPad Software, San Diego, California USA, www.graphpad.com). Demographics variables were presented as medians and interquartile ranges. The statistical raw data for blood oxygen indices is presented as means, as they were normally distributed for day one to 30 of admission. For demographic comparisons between survivors and non-survivors, we used Mann-Whitney U test for continuous variables and Fisher's Exact test for categorical data. For blood oxygen indices with normal distribution, we used Welch-two sample t-test to compare survivors and non-survivors. Pearson's correlation coefficient was used to assess the relationship between individual blood oxygen indices and haemoglobin. The accuracy of individual blood oxygen indices in predicting mortality was assessed using area under receiver operating curves (AUROC). Median values are presented with the interquartile range (IQR), mean values are presented with confidence intervals (95% CI) and categorical data are presented with percentage (%). Statistical significance was assumed when p value of <0.05. We used the Benjamini-Hochberg (BH) adjustment to reduce the false discovery (type I error) rate when performing multiple statistical tests [16,17].

## Results

During this study period, there were 1835 SARS-CoV-2 positive hospital admissions, of which 340 required admissions to the critical care unit. 184 patients required invasive mechanical ventilation, providing a total of just over 36,383 serial arterial or venous blood gas data points over the course of the first 30 days, all of which were included in the analysis (Fig 1). For these patients received invasive mechanical ventilation, the 30-day hospital survival rate was 72.3%. Baseline demographic, laboratory and ICU interventions and outcomes of all patients and comparison between survivors and non-survivors are presented in Table 1.

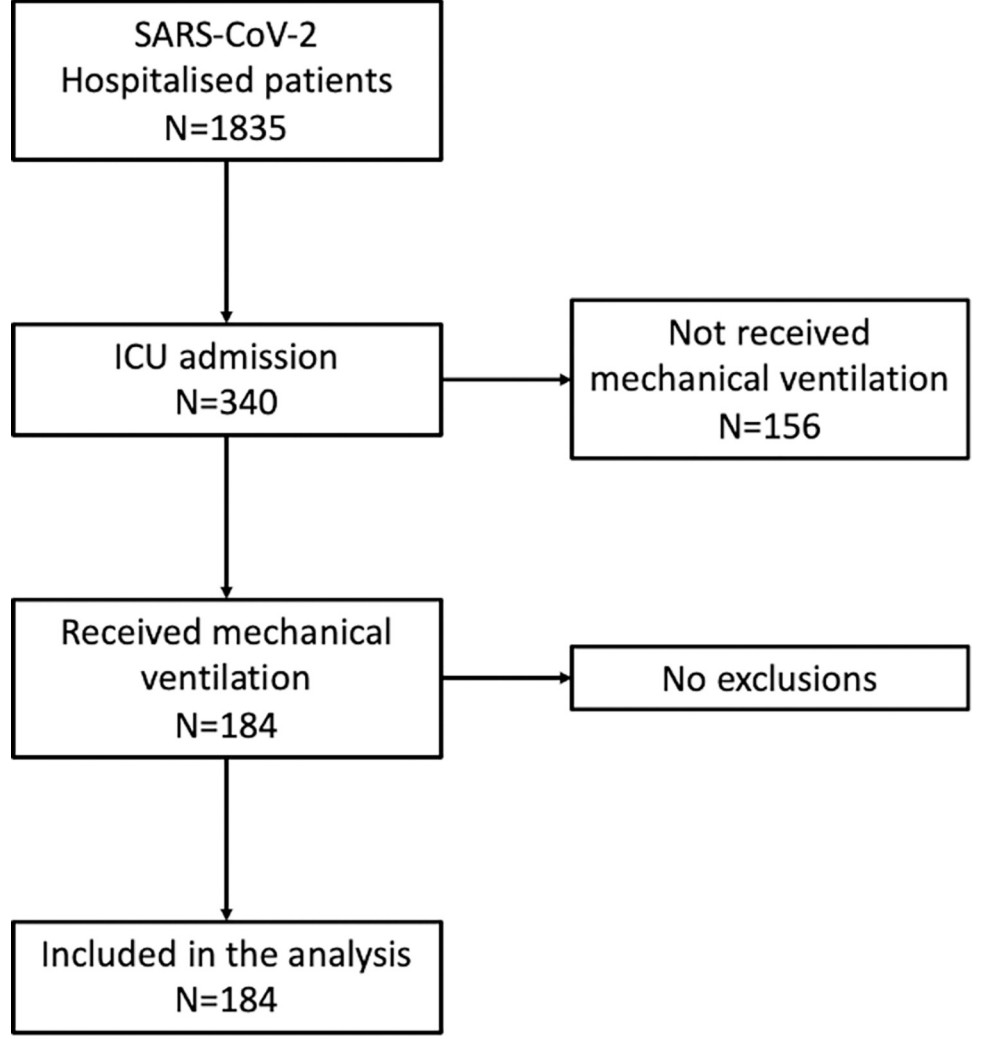

**Fig 1. Flow diagram of included participants.**

The median age was 59.5 years (IQR 51.0, 67.0), and survivors were significantly younger, 57.0 (IQR 49.0, 64.0) compared to non-survivors, 65.0 (IQR 59.5, 72.0) with male predominance (62.5%). Gender was not associated with increased mortality. 129 patients were of white ethnic origin (70.1%), and 55 were from ethnic minority groups (29.9%). The median body mass index (BMI) was 30.0 (IQR 25.8, 35.5), with no significant difference between survivors, 30.1 (IQR 25.2, 35.8) and non-survivors 29.4 (IQR 26.6, 33.8). The median admission Acute Physiology and Chronic Health Evaluation II (APACHE II) score was 18 (IQR 12, 23) giving a predictive mortality of 25%. Survivors had a significantly lower score, 16.0 (IQR 11.0, 23.0) compared to non-survivors, 19.0 (IQR 14.3, 24.8) both of which also fall into the 25% expected mortality prediction. The median Sequential Organ Failure Assessment (SOFA) score on admission was 4.0 (IQR 3.0, 7.0) giving a predictive mortality of 36.1%. There was no significant difference in the SOFA score between survivors, 4.0 (IQR 3.0, 6.0) and non-survivors, 5.5 (IQR 4.0, 8.0). The median Clinical Frailty Score (CFS) was 2.0 (IQR 2.0, 3.0) meaning 'well', with no significant difference between survivors, 2.0 (IQR 2.0, 3.0) and non-survivors, 2.0 (IQR 2.0, 4.0). Median Charlson Comorbidity Index (CCI) was 2.0 (IQR 1.0, 3.0), with

**Table 1. Patient characteristics and outcomes of all patients meeting inclusion criteria (n = 184).**

| Variables | All patients N = 184 | Survivors N = 133 | Non-survivors N = 51 | p-value |
|---|---|---|---|---|
| **Age** | 59.5 (51.0, 67.0) | 57.0 (49.0, 64.0) | 65.0 (59.5,72.0) | <0.01 |
| **Sex, n (%)** | | | | |
| Male | 115 (62.5%) | 79 (59.4%) | 36 (70.6%) | 0.18 |
| Female | 69 (37.5%) | 54 (40.6%) | 15 (29.4%) | |
| **BMI (kg/m²)** | 30.0 (25.8, 35.5) | 30.1 (25.2, 35.8) | 29.4 (26.6, 33.8) | 0.75 |
| **Ethnicity, n (%)** | | | | |
| White | 129 (70.1%) | 95 (71.4%) | 34 (66.7%) | 0.59 |
| Asian | 31 (16.8%) | 21 (15.8%) | 10 (19.6%) | 0.52 |
| Black | 14 (7.6%) | 10 (7.5%) | 4 (7.8%) | 1.0 |
| Mixed | 6 (3.3%) | 5 (3.8%) | 1 (2.0%) | 1.0 |
| Unknown | 4 (2.2%) | 2 (1.5%) | 2 (3.9%) | 0.31 |
| **Clinical Frailty Score** | 2.0 (2.0, 3.0) | 2.0 (2.0, 3.0) | 3.0 (2.0, 4.0) | 0.06 |
| **Charlson Comorbidity Index** | 2.0 (1.0, 3.0) | 2.0 (1.0, 3.0) | 3.0 (3.0, 4.5) | <0.01 |
| **Comorbidities, n (%)** | | | | |
| Asthma | 19 (10.3%) | 12 (9.0%) | 7 (13.7%) | 0.42 |
| Chronic obstructive pulmonary disease | 11 (6.0%) | 6 (4.5%) | 5 (9.8%) | 0.18 |
| Diabetes mellitus | 56 (30.4%) | 40 (30.1%) | 16 (31.4%) | 0.86 |
| Hypertension | 78 (42.4%) | 57 (42.9%) | 21 (41.1%) | 0.87 |
| Ischaemic heart disease | 16 (18.7%) | 7 (5.3%) | 9 (17.6%) | 0.02 |
| Chronic kidney disease | 11 (3.3%) | 5 (3.8%) | 6 (11.8%) | 0.07 |
| Immunosuppression | 22 (12.0%) | 16 (12.0%) | 8 (15.7%) | 0.63 |
| **Admission arterial blood gas** | | | | |
| pH | 7.44 (7.38, 7.48) | 7.44 (7.40, 7.48) | 7.43 (7.35, 7.48) | 0.22 |
| $PaO_2$ (kPa) | 9.4 (8.5, 11.1) | 9.3 (8.4, 11.5) | 9.6 (8.7, 10.5) | 0.98 |
| $PaCO_2$ (kPa) | 5.0 (4.5, 5.9) | 5.0 (4.5, 5.7) | 5.0 (4.5, 6.5) | 0.58 |
| $PaO_2/FiO_2$ | 15.0 (12.1, 19.1) | 15.0 (12.3, 18.9) | 15.4 (12.1, 19.7) | 0.51 |
| $HCO_3^-$ (mmol/L) | 25.7 (23.2, 28.1) | 25.9 (23.9, 28.2) | 24.9 (20.7, 27.2) | 0.01 |
| Base excess (nmol/L) | 1.4 (-1.0, 3.6) | 1.6 (-0.1, 3.8) | 1.2 (-3.2, 2.9) | 0.03 |
| Lactate (mmol/L) | 1.2 (0.9, 1.6) | 1.1 (0.8, 1.5) | 1.4 (1.0, 1.8) | 0.01 |
| **Admission lab variables** | | | | |
| Bilirubin (μmol/L) | 11 (8, 14) | 11 (7, 15) | 10 (8, 13) | 0.93 |
| Creatinine (μmol/L) | 73 (55, 98) | 68 (51, 96) | 84 (66, 106) | 0.01 |
| Creatinine kinase (IU/L) | 128 (57, 386) | 132 (64, 393) | 99 (54, 329) | 0.75 |
| CRP (mg/L) | 125 (67, 192) | 133 (77, 192) | 97 (51, 184) | 0.29 |
| D-Dimer (μg/L) | 619 (340, 1283) | 614 (324, 1148) | 667 (358, 2017) | 0.31 |
| Ferritin (μg/L) | 687 (381, 1168) | 656 (373, 1093) | 871 (543, 1339) | 0.14 |
| INR | 1.1 (1.0, 1.2) | (1.0, 1.2) | (1.0, 1.2) | 0.25 |
| LDH (IU/L) | 968 (760, 1276) | 910 (755, 1231) | 1098 (829, 1431) | 0.05 |
| Lymphocytes (x10⁹/L) | 0.7 (0.5, 1.0) | 0.7 (0.5, 1.0) | 0.7 (0.5, 0.9) | 0.70 |
| Neutrophil/lymphocyte ratio | 10.5 (6.7, 18) | 10.3 (6.1, 17.5) | 10.6 (6.8, 19.6) | 0.58 |
| Procalcitonin (ng/L) | 0.3 (0.1, 1.0) | 0.3 (0.1, 0.9) | 0.3 (0.1, 1.0) | 0.67 |
| Troponin (ng/L) | 15 (9, 53) | 15 (8, 37) | 20 (10, 71) | 0.11 |
| **ICU severity scores on admission** | | | | |
| APACHE II | 18.0 (12.0, 23.0) | 16.0 (11.0, 23.0) | 19.0 (14.3, 24.8) | 0.01 |
| SOFA | 4.0 (3.0, 7.0) | 4.0 (3.0, 6.0) | 5.5 (4.0, 8.0) | 0.09 |
| **ICU interventions** | | | | |
| Pre-intubation NIV/CPAP, n (%) | 113 (61.4%) | 93 (69.9%) | 32 (62.7%) | 0.38 |
| Prone positioning, n (%) | 147 (79.9%) | 109 (82.0%) | 38 (74.5%) | 0.30 |
| Renal replacement therapy, n (%) | 50 (27.2%) | 30 (22.6%) | 20 (39.2%) | 0.03 |
| **Duration of mechanical ventilation (days)** | 15 (8, 25) | 16 (11, 29) | 8 (5, 15) | <0.01 |
| **Duration of ICU length of stay (days)** | 20 (11, 36) | 24 (17, 42) | 11 (6, 17) | <0.01 |
| **Duration of hospital length of stay (days)** | 28 (19, 53) | 41 (27, 64) | 16 (10, 20) | <0.01 |

All scores and laboratory variables were performed at the time of ICU admission. APACHE II: Acute physiology and chronic health evaluation; BMI: Body mass index; CRP: C-Reactive protein; ICU: Intensive care unit; INR: International normalised ratio; LDH: Lactate dehydrogenase; SOFA: Sequential organ failure assessment.

survivors scoring significantly lower, 2.0 (IQR 1.0, 3.0) classed as 'moderate', compared to non-survivors, 3.0 (IQR 3.0, 4.5) classed as 'severe'. Presence of ischaemic heart disease was associated with non-survival (Table 1). Differences between survivors and non-survivors for other comorbidities (asthma, chronic obstruction pulmonary disease, diabetes, hypertension, chronic kidney disease and pre-existing immunosuppression) were not significant.

Median duration of mechanical ventilation was 15 days (IQR 8, 25). Survivors were mechanically ventilated for significantly longer, 16 days (IQR 11, 29) compared to non-survivors, 8 days (IQR 5, 15). Median duration of ICU length of stay was 20 days (IQR 11, 36). Survivors stayed on ICU for significantly longer, 24 days (IQR 17, 42) compared to non-survivors, 11 days (IQR 6, 17). Median duration of hospital stay was 28 days (IQR 19, 53), with survivors staying significantly longer, 41 days (IQR 27, 64), compared to non-survivors, 16 days (IQR 10, 20). 113 patients (61.4%) received non-invasive ventilation prior to intubation, with no significant difference between survivors, 93 (69.9%), and non-survivors, 32 (62.7%). 147 patients (79.9%) received prone positioning as part of their care, and there was no significant difference between survivors, 109 (82.0%), and non-survivors, 38 (74.5%). 50 patients (27.2%) required renal replacement therapy, with a significant difference between survivors, 30 (22.6%) and non-survivors, 20 (39.2%).

Median admission lactate was 1.2 (IQR 0.9, 1.6) and was significantly lower in survivors, 1.1 (IQR 0.8, 1.5) than non-survivors, 1.4 (IQR 1.0, 1.8). Median admission $HCO_3^-$ was 25.7 (IQR 23.2, 28.2) and was significantly higher in survivors, 26.2 (IQR 23.9, 28.2) than non-survivors, 24.9 (IQR 21.9, 27.2). Median admission base excess was 1.4 (IQR -1.0, 3.6) and was significantly higher in survivors, 1.6 (IQR -0.1, 3.8), than non-survivors, 1.2 (IQR -3.2, 2.9). There was no other significant difference in admission arterial blood gas values. Admission creatinine was 73 (IQR 55, 98) and was significantly lower in survivors, 68 (IQR 51, 96) than non-survivors, 84 (IQR 66, 106). There were no other significant differences in baseline admission laboratory blood results between survivors and non-survivors. Detailed patient's demographics and outcomes are presented in Table 1.

Arterial oxygen indices from 34,592 sampling points across days one to seven and days one to 30 are detailed in Table 2.

Non-survivors exhibited significantly lower $PaO_2$ throughout the admission (Fig 2A). From day one to day seven of ICU admission there was an averaged mean difference of -0.31 kPa (95% CI: -0.41 to -0.20) and from day one to day 30 an averaged mean difference of -0.52 kPa

**Table 2. Comparison of mean averaged blood oxygen indices at day one-seven, and day one-30; survivors (n = 133) vs. non-survivors (n = 51).**

|  | Survivors | Non-survivors | Mean Difference | 95% CI | p-value* |
|---|---|---|---|---|---|
| **$PaO_2$ (kPa)** |  |  |  |  |  |
| Day 1–7 | 9.80 | 9.49 | -0.31 | -0.41, -0.20 | <0.01 |
| Day 1–30 | 9.73 | 9.21 | -0.52 | -0.59, -0.46 | <0.01 |
| **$PaO_2$ (kPa) / $FiO_2$ ratio** |  |  |  |  |  |
| Day 1–7 | 19.74 | 17.51 | -2.23 | -2.55, -1.91 | <0.01 |
| Day 1–30 | 21.19 | 15.56 | -5.64 | -5.85, -5.43 | <0.01 |
| **$CaO_2$ (ml/dL)** |  |  |  |  |  |
| Day 1–7 | 14.33 | 14.63 | 0.31 | 0.19, 0.42 | <0.01 |
| Day 1–30 | 12.78 | 13.62 | 0.83 | 0.75, 0.91 | <0.01 |
| **$O_2ER$** |  |  |  |  |  |
| Day 1–7 | 0.34 | 0.27 | -0.07 | -0.09, -0.04 | <0.01 |
| Day 1–30 | 0.38 | 0.29 | -0.08 | -0.09, -0.07 | <0.01 |

* Using the Benjamini-Hochberg adjustment.

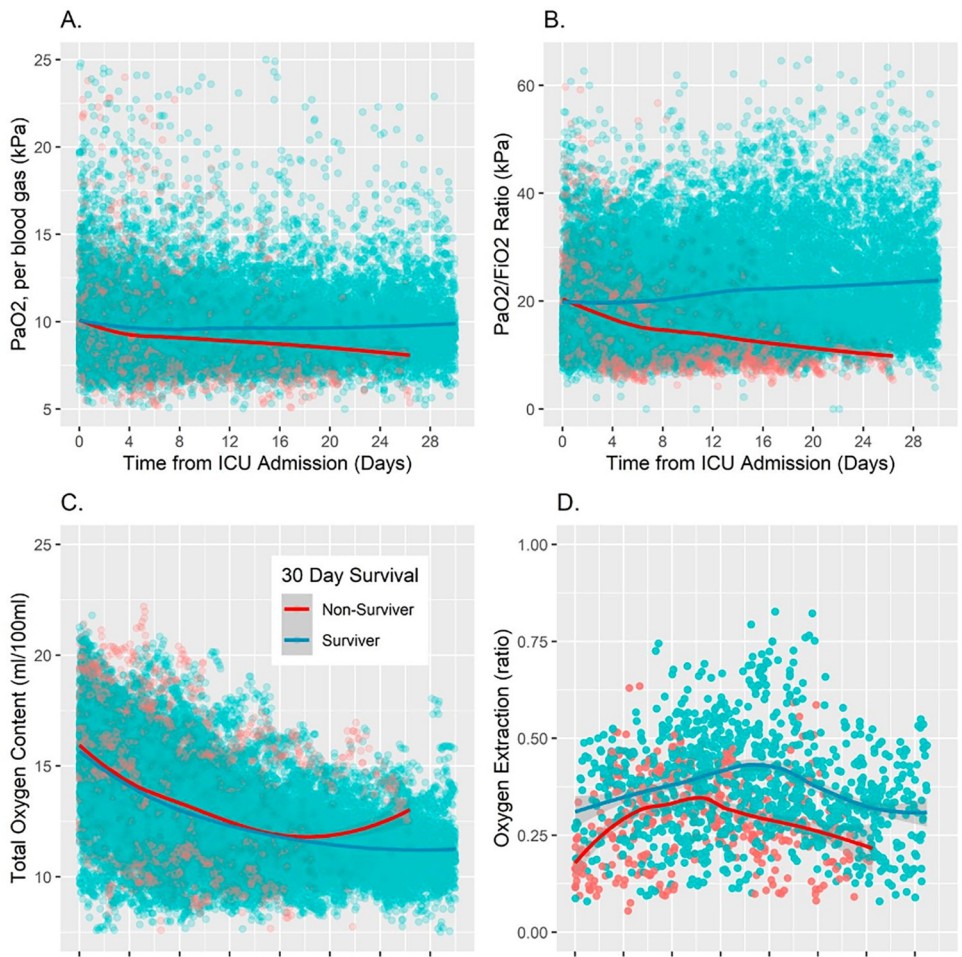

**Fig 2. Blood oxygen indices over time between survivors and non-survivors.** (A) $PaO_2$ (kPa), (B). $PaO_2$ / $FiO_2$ ratio (kPa), (C). Total oxygen content (ml/dL), (D). Oxygen extraction ratio ($O_2ER$).

(95% CI: -0.59 to -0.46). Moreover, non-survivors exhibited a significantly lower $PaO_2/FiO_2$ ratio, with improved separation over time (Fig 2B). Across day one to day seven of ICU admission there is an averaged mean difference of -2.23 (95% CI: -2.55 to -1.91) and across day one to day 30 an averaged mean difference of -5.64 (95% CI: -5.85 to -5.43). While both survivors and non-survivors exhibited sub-physiological $CaO_2$ (trending down throughout admission, survivors exhibited significantly lower values (Fig 2C). Across day one to day seven of ICU admission there is an averaged mean difference in $CaO_2$ of 0.31 (95% CI: 0.19 to 0.42) and for day one to day 30 an averaged mean difference of 0.83 (95% CI 0.75: to 0.91). For oxygen extraction analysis (Table 2), 1,791 data points were available with contemporaneous arterial and venous blood sampling. Non-survivors exhibited significantly lower oxygen extraction (Fig 2D). From day one to day seven of ICU admission there was an averaged mean difference in $O_2ER$ of -0.07 (95% CI: -0.09 to -0.04) and from day one to day 30 an averaged mean difference of -0.08 (95% CI: -0.09 to -0.07). As expected, there was a tight, linear correlation between $CaO_2$ and haemoglobin concentrations (Fig 3).

We studied the use of these blood oxygen indices in predicting hospital mortality of all mechanically ventilated patients. The area under the receiver operating characteristic curve (AUC) for $FiO_2$ and $PaO_2/FiO_2$ were similar at 0.78 (95% CI: 0.77 to 0.79, p<0.01) and 0.78

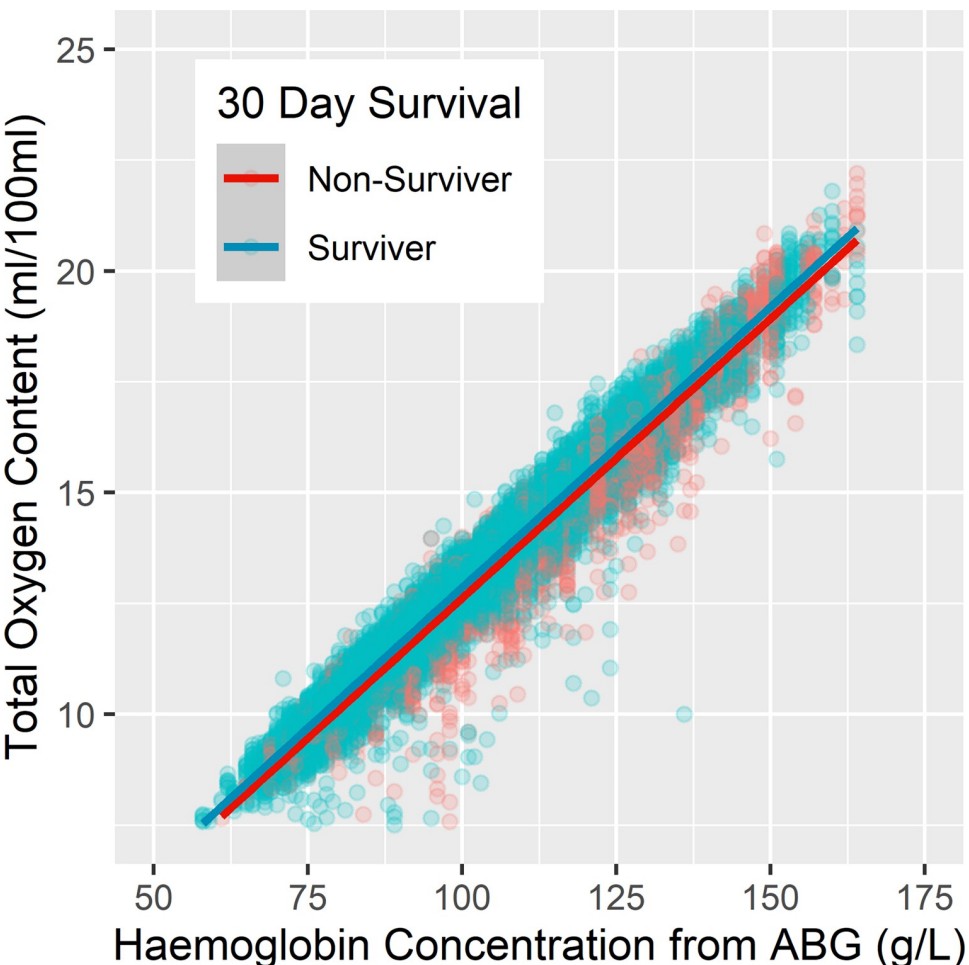

**Fig 3. The correlation between oxygen content and haemoglobin.**

(95%CI: 0.77 to 0.78, p<0.01) with a cut-off value of $FiO_2 > 0.54$ and $PaO_2/FiO_2 \leq 16.6$ kPa of respectively (Fig 4A). While $PaO_2$ was less predictive with an AUC of 0.60 (95% CI: 0.60 to 0.61, p< 0.01, cut-off 9.3 kPa) (Fig 4A). $O_2ER$ was a better predictor of hospital mortality than $CaO_2$ with an AUC of 0.70 (95%CI: 0.67 to 0.72, p<0.01, cut off $\leq$0.29) (Fig 4B and 4C).

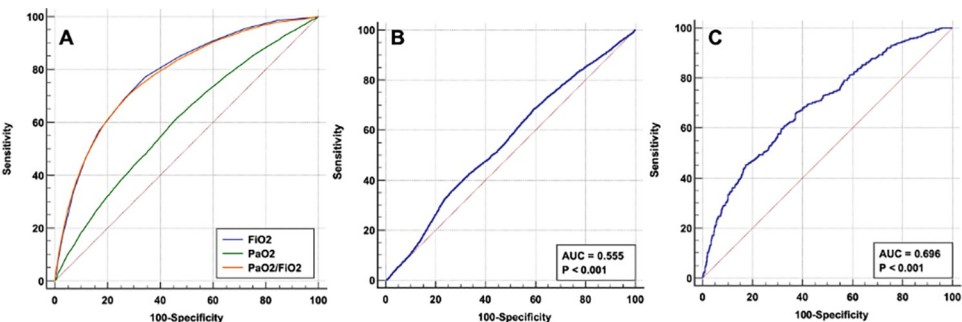

**Fig 4. Blood oxygen indices operating characteristics analysis.** (A). AUC for $FiO_2$, $PaO_2$ and $PaO_2/FiO_2$, (B). Total oxygen content ($CaO_2$), (C). Oxygen extraction ratio ($O_2ER$).

Although statistically significant, total oxygen content was less predictive of hospital mortality than other blood oxygen indices (Fig 4).

## Discussion

In this observational study, we have demonstrated that there were significant abnormalities in blood oxygen indices in mechanically ventilated adult COVID-19 patients. Of note, despite having higher total oxygen content, non-survivors exhibited lower oxygen extraction ratios. These findings support the notion that mechanically ventilated adult patients with COVID-19 may have impaired oxygen utilisation and that this is a marker of severity of disease.

The physiology of oxygen transport is well-described [15]. High-quality evidence to support the optimal measure of oxygenation in critically unwell patients is limited and most research has tended to consider $SaO_2$, $PaO_2$ or $PaO_2/FiO_2$ ratios in isolation and are conducted in heterogenous ICU cohorts with various underlying pathologies. Oxygenation targets for patients admitted to intensive care are conflicting [18], and while it appears in general that over-oxygenation might be harmful [19], diverse groups of patients with differing pathologies are unlikely to all benefit from a single approach. Some COVID-19 patients are at risk of profound hypoxemic respiratory failure and development of acute respiratory distress syndrome (ARDS), with a number of mechanisms proposed including intrapulmonary shunting, impaired lung perfusion regulation, intravascular microthrombi and impaired diffusion capacity at a tissue level [5]. Despite degree of hypoxaemia being predictive of mortality, oxygen targets for these patients are not yet well-established [20], though correction of oxygenation may improve survival [21] and some are calling for higher as well as lower targets [22].

Arterial partial pressure of oxygen ($PaO_2$), though commonly used, gives no indication of the required inspired oxygen. Additional information from $PaO_2/FiO_2$ ratios can provide further indication as to the degree of hypoxic respiratory failure. These ratios are helpful in stratifying the severity of ARDS as part of the Berlin Definition [23], but are dependent on the operator input of correct $FiO_2$ and temperature [24,25]. In patients with severe hypoxic respiratory failure, the $PaO_2/FiO_2$ ratio is often the primary blood oxygen index used to guide decisions regarding initiation of mechanical ventilation, escalation of ventilatory support or when to institute rescue measures such as prone positioning and extracorporeal membrane oxygenation (ECMO). In our patient cohort, the $PaO_2/FiO_2$ ratio continued to deteriorate over the 7-day period in non-survivors and was a better predictor of ICU survival in all mechanically ventilated COVID-19 patients than other blood oxygen indices. Despite a statistical difference, the association between $PaO_2/FiO_2$ ratio and the $CaO_2$ was weak, suggesting that an increment in fractional inspired oxygen may not correspondingly influence oxygen content in all patients. Moreover, although oxygen content was low overall, there was relative preservation in non-survivors with no difference between the groups.

The commonly measured blood oxygen indices ($SaO_2$, $PaO_2$) may quantify the degree of COVID-19-related respiratory failure, but may not inform on oxygen delivery to the tissues. Recent studies have failed to demonstrate a survival benefit from optimisation of oxygen delivery ($DO_2$) by the manipulation of supplemental oxygen, blood volume expansion, and cardiovascular supportive measures in sepsis [26–30], however as previously observed these studies represent more heterogenous underlying pathologies than COVID-19. In health, oxygen uptake ($VO_2$) is well-maintained even with a decreasing $DO_2$ due to a variety of compensatory mechanisms including increased $O_2ER$ and redistribution of blood flow to tissues with the highest oxygen demand. It has been suggested that $VO_2$ decreases below a so-called "critical $DO_2$ ($DO_2crit$)", where $O_2ER$ is maximal. Tissue hypoxia may occur if $DO_2$ continues to decrease below a notional $DO_2crit$, or if $VO_2$ increases or fails, resulting in anaerobic

respiration, lactate production and ultimately, ischaemia. This situation may be exacerbated by fever, rigors and sepsis [31], all features of severe COVID-19 infection.

Examination of $SvO_2$ may indicate the balance between $DO_2$ and $VO_2$, where a value of >70% suggests adequate respiration. In severe sepsis, where tissue level oxygen metabolism is impaired due to microcirculation disorders and inflammatory mediator damage, a lower $O_2ER$ is strongly associated with increased mortality [32]. Our findings suggest $O_2ER$ is statistically lower in non-survivors than survivors. This finding contradicts a recent study which suggested an increased $O_2ER$ in non-survivors of severe COVID-19 infection [7]. However, this work is not directly comparable to our study as the authors only used admission blood sample data for calculations, to provide a 'snapshot' of admission parameters. It is unclear what proportion required respiratory support in this study, in contrast to our study in which all patients were mechanically ventilated. Moreover, the COVID-19 survivors had lower $O_2ER$ than healthy controls and the method of $O_2ER$ estimation was also different from our study by including estimates of cardiac output. Similar to our study, a small case series also reported reduced oxygen utilisation in patients with COVID-19 which may be associated with adverse outcomes [33].

The reasons for our finding of association of lower $O_2ER$ with mortality is uncertain, though are likely to be multi-factorial. Up-regulation of $O_2ER$ when $DO_2$ is reduced may fail in severe pathology such as tissue hypoxia or acidosis, though curiously $O_2ER$ is not increased in healthy, acclimatised individuals at altitude (i.e., hypobaric hypoxia) [34], possibly due to hypoxia itself reducing the ability to extract oxygen. It has been documented that COVID-19 is implicated with a multi organ microangiopathic process with endotheliopathy, vascular thrombosis, overt inflammatory cytokine response and abnormalities of von Willebrand factor-platelet axis [35,36]. The interaction between virus and receptor is thought to downregulate ACE2 activation, thus increasing levels of angiotensin II with consequences of intense vasoconstriction, inflammation and oxidative stress enhancing thrombogenicity. The cumulative effect of both endotheliopathy and vasoconstriction may contribute to a scenario with tissue level reduced oxygen utilisation. This may explain our findings that a lack of ability to upregulate oxygen extraction led to poorer outcomes. Another most important consideration for our findings could be mitochondrial dysfunction [37]. While 90% of total oxygen consumption is accounted for by mitochondrial oxidative phosphorylation, mitochondrial function extends beyond adenosine triphosphate (ATP) production, playing critical roles in cellular messaging, apoptosis, autophagy, and calcium homeostasis [38]. There is increasing evidence that upon cell entry the SARS-CoV-2 virus hijacks host's mitochondria which may contribute to mitochondrial dysfunction and cellular death [39,40]. The associations of increased mortality with advanced age, metabolic syndrome and immune deficiency may reflect existing mitochondrial dysfunction among these groups exposing their vulnerability [41].

To our knowledge this is the first study to establish the dynamic trend of $PaO_2$, $PaO_2/FiO_2$, $O_2ER$ and $CaO_2$ over the course of an ICU admission in patients with severe COVID-19 pneumonia.

Limitations include a retrospective design, with no prior power calculation, and potentially our significant results are the result of a type II error. Our sample size is reasonable, but only 51 patients died. This could lead to non-survivors being under-represented, though we tried to address this by only including intubated patients. Our data is clearly susceptible to sampling bias, as the sickest patients tend to get more frequent blood gas sampling than more stable patients. However, our results show that those who died were not sampled excessively when compared with non-survivors. The blood oxygen indices we describe are estimates for overall gas analysis, not for individual patients. Additional analysis designed to reduce this bias

yielded similar results suggesting a degree of robustness. In addition, we analysed patients mechanically ventilated at any point, including those who were transferred to our centre under mutual aid, transferred out for ECMO, and those for who incidentally positive for COVID-19, but initially admitted for other pathology. Moreover, extraction ratios were calculated with $ScvO_2$ from central venous lines, than $SvO_2$ samples from pulmonary artery catheters. While $ScvO_2$ correlates well with $SvO_2$, it essentially reflects the oxygenation of the upper body and head, not including myocardial perfusion. This may be important for patients with COVID-19 due to cardiomyopathy and relatively high-output cardiac states.

## Conclusions

The COVID-19 pandemic offers a unique opportunity to study a homogenous cohort of hypoxaemic critically unwell patients, with similar underlying pathology. In a cohort of mechanically ventilated adult ICU patients with hypoxaemic respiratory failure due to COVID-19, oxygen extraction is significantly lower in non-survivors compared to survivors during the first 30 days of ICU admission, despite having higher $CaO_2$ values. This suggests COVID-19 may cause impaired oxygen utilisation. Urgent further evaluation of the relationship between impaired oxygen extraction and survival in COVID-19 is justified.

## Acknowledgments

We thank all the members of the REACT study group

Tom Wilkinson (University of Southampton) T.Wilkinson@soton.ac.uk Lead collaborator.

Anna Freeman (University of Southampton)

Hannah Burke (University of Southampton)

Michael Celinski (University Hospital Southampton)

Saul N Faust (University of Southampton)

Gareth J Thomas (University of Southampton)

Christopher Kipps (University of Southampton)

## Author Contributions

**Conceptualization:** Luke Bracegirdle, Andrew Cumpstey, Ahilanandan Dushianthan.

**Data curation:** Luke Bracegirdle, Alexander Jackson, Ryan Beecham, Maria Burova, Elsie Hunter, Laura G. Hamilton, Darshni Pandya, Clare Morden, Ahilanandan Dushianthan.

**Formal analysis:** Luke Bracegirdle, Alexander Jackson, Ahilanandan Dushianthan.

**Investigation:** Ahilanandan Dushianthan.

**Methodology:** Luke Bracegirdle, Alexander Jackson, Maria Burova, Elsie Hunter, Laura G. Hamilton, Darshni Pandya, Clare Morden, Andrew Cumpstey, Ahilanandan Dushianthan.

**Project administration:** Luke Bracegirdle, Ahilanandan Dushianthan.

**Supervision:** Michael P. W. Grocott, Ahilanandan Dushianthan.

**Writing – original draft:** Luke Bracegirdle, Andrew Cumpstey, Ahilanandan Dushianthan.

**Writing – review & editing:** Luke Bracegirdle, Alexander Jackson, Ryan Beecham, Maria Burova, Elsie Hunter, Laura G. Hamilton, Darshni Pandya, Clare Morden, Michael P. W. Grocott, Andrew Cumpstey, Ahilanandan Dushianthan.

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
