## [Decision Letter · Decision Letter 0]

7 Mar 2022

PONE-D-21-30166Dynamic blood oxygen indices in mechanically ventilated COVID-19 patients with acute hypoxic respiratory failure: a cohort studyPLOS ONE

Dear Dr. Grocott,

Thank you for submitting your manuscript to PLOS ONE. After careful consideration, we feel that it has merit but does not fully meet PLOS ONE’s publication criteria as it currently stands. Therefore, we invite you to submit a revised version of the manuscript that addresses the points raised during the review process.

We look forward to receiving your revised manuscript.

Kind regards,

Antonino Salvatore Rubino, M.D., Ph.D.

Academic Editor

PLOS ONE

“The authors received no speicifc fubnding for this work.”

d) If you did not receive any funding for this study, please state: “The authors received no specific funding for this work.

4. One of the noted authors is a group or consortium Tom Wilkinson, Anna Freeman, Hannah Burke, Michael Celinski, Saul N Faust, Gareth J Thomas, Christopher Kipps. In addition to naming the author group, please list the individual authors and affiliations within this group in the acknowledgments section of your manuscript. Please also indicate clearly a lead author for this group along with a contact email address.

Reviewers' comments:

Reviewer's Responses to Questions

**Comments to the Author**

1. Is the manuscript technically sound, and do the data support the conclusions?

Reviewer #1: Yes

Reviewer #2: No

2. Has the statistical analysis been performed appropriately and rigorously? 

Reviewer #1: Yes

Reviewer #2: No

3. Have the authors made all data underlying the findings in their manuscript fully available?

Reviewer #1: Yes

Reviewer #2: Yes

4. Is the manuscript presented in an intelligible fashion and written in standard English?

Reviewer #1: Yes

Reviewer #2: Yes

5. Review Comments to the Author

Reviewer #1: Your work is applaudable. Please find below some comments and suggestions:

1. "Due to the nature of this study, the need for individual informed patient consent was waived." Can the authors provide further explanation as to why the waiver is applicable to this particular study?

2. "We did not exclude any patients based on the presence of co-morbidities that may have contributed to their death or those enrolled in other clinical trials." Can the authors provide further explanation as to why such patients were not within the frames of an exclusion criteria?

3. Can the authors provide their limitation in a separately headed segment? Or present it separately from the first statement of the paragraph that highlights limitation?

4. Can the authors provide statements in their conclusion that highlight the importance of their findings in terms of shaping clinical practice or global/public health practices?

Reviewer #2: This retrospective cohort study evaluated relationship between three oxygen indices and clinical outcomes in mechanically ventilated COVID-19 patients. Although concept of the study is reasonable and the manuscript is well written, methods have serious problem.

Major points:

1) As authors mentioned in a limitation part (P22, L364), the results of this study are affected by serious sampling bias, which has bearings on the fundamental point of the study design. Including all blood samples into the analysis cannot be reasonable, since more samples from sicker patients must lead to patently false conclusion. I totally agree with authors opinion that oxygen extraction rate is associated with mortality of COVID-19 patients, but I cannot agree this study is technically sound and appropriate, as long as they use quite biased samples. Since this study included 181 patients, number of blood gas samples must be multiples of 181.

Minor points:

2) P21, L353-357: Discussion about mitochondrial dysfunction looks too strong and too assertive, even though I agree with the opinion that mitochondrial function should affect oxygen utilization and clinical outcome of this patient population.

3) P22, L381-382: The same as above.

6. PLOS authors have the option to publish the peer review history of their article (what does this mean?). If published, this will include your full peer review and any attached files.

Reviewer #1: No

Reviewer #2: No

---

## [Author Response · Author response to Decision Letter 0]

20 Apr 2022

Thank you for the valuable comments. Please see our response to the reviewer’s comments. 

Reviewer #1: 

Your work is applaudable. Please find below some comments and suggestions:

1. "Due to the nature of this study, the need for individual informed patient consent was waived." Can the authors provide further explanation as to why the waiver is applicable to this particular study?

Response: Thank you for your comment. We have now modified this sentence to clarify this further as “due to the retrospective and observational nature of the study and there were no identifiable patient’s source data, the need for individual informed patient consent was waived. The data analysed were already routinely collected and electronically stored as part of clinical care”

2. "We did not exclude any patients based on the presence of co-morbidities that may have contributed to their death or those enrolled in other clinical trials." Can the authors provide further explanation as to why such patients were not within the frames of an exclusion criteria?

Response: Thank you. We have now modified this sentence to clarify this statement further. “As a pragmatic retrospective study without intervention, we evaluated the merit of various oxygen indices under a real-life, generalised intensive care setting which may be applicable to routine practice. We therefore did not exclude any patients based on the presence of comorbidities that may have contributed to their death or those enrolled in other clinical trials”.

3. Can the authors provide their limitation in a separately headed segment? Or present it separately from the first statement of the paragraph that highlights limitation?

Response: We have now presented the limitations as a separate paragraph. 

4. Can the authors provide statements in their conclusion that highlight the importance of their findings in terms of shaping clinical practice or global/public health practices?

Response: Thank you. While this is an important clinical finding that oxygen utilisation may be compromised in severe fatal COVID-19 patients, this warrants further clinical and scientific explorations to shape any future clinical practice (i.e., measure to improve oxygen utilisation, is not straight forward). This a pragmatic observational study suggests that improving oxygen content alone may not be adequate. Any additional comments on change in clinical practice would be exploratory. 

Reviewer #2: 

This retrospective cohort study evaluated relationship between three oxygen indices and clinical outcomes in mechanically ventilated COVID-19 patients. Although concept of the study is reasonable and the manuscript is well written, methods have serious problem.

Major points:

1) As authors mentioned in a limitation part (P22, L364), the results of this study are affected by serious sampling bias, which has bearings on the fundamental point of the study design. Including all blood samples into the analysis cannot be reasonable, since more samples from sicker patients must lead to patently false conclusion. I totally agree with authors opinion that oxygen extraction rate is associated with mortality of COVID-19 patients, but I cannot agree this study is technically sound and appropriate, as long as they use quite biased samples. Since this study included 181 patients, number of blood gas samples must be multiples of 181.

Response: Thank you for your comment. This was a pragmatic observational study and consequently, we were unable to prescribe the timing and quantity of oxygen measurements that is suitable for an individual patient in ICU. We have clarified this in the limitations section. While we agree that might be variations in sampling episodes between patients and would have introduced precision bias, this is highly unlikely as the sampling episodes were similar between survivors and non-survivors (Table). Moreover, all the patients presented here were very sick and required mechanical ventilation, representing a unique homogenous cohort with similarities in their degree of hypoxia. We presented all available blood samples to minimise any selection bias. We have modified the sentence in the limitations to address this comment. “Our data is clearly susceptible to sampling bias, as the sickest patients tend to get more frequent blood gas sampling than more stable patients. However, our results show that those who died were not sampled excessively when compared with non-survivors”.

Minor points:

2) P21, L353-357: Discussion about mitochondrial dysfunction looks too strong and too assertive, even though I agree with the opinion that mitochondrial function should affect oxygen utilization and clinical outcome of this patient population.

Response: We have modified this sentence to reflect the reviewer’s comments. “There is increasing evidence that upon cell entry the SARS-CoV-2 virus hijacks host's mitochondria which may contribute to mitochondrial dysfunction and cellular death”. 

3) P22, L381-382: The same as above.

Response: We have modified the sentence to reflect the reviewer’s comments. “Urgent further evaluation of the relationship between impaired oxygen extraction and survival in COVID-19 is justified”.

The changes within the manuscript are highlighted in yellow. We are looking forward to hearing from you soon. 

Thank you for considering this manuscript for publication in PLOS one. 

Yours sincerely,

Professor Mike Grocott 

BSc MBBS MD FRCA FRCP FFICM

Professor of Anaesthesia and Critical Care Medicine, Head, Integrative Physiology and Critical Illness Group, CES Lead, Critical Care Research Area, Southampton NIHR Respiratory BRC

---

## [Decision Letter · Decision Letter 1]

23 May 2022

Dynamic blood oxygen indices in mechanically ventilated COVID-19 patients with acute hypoxic respiratory failure: a cohort study

PONE-D-21-30166R1

Dear Dr. Grocott,

We’re pleased to inform you that your manuscript has been judged scientifically suitable for publication and will be formally accepted for publication once it meets all outstanding technical requirements.

Kind regards,

Antonino Salvatore Rubino, M.D., Ph.D.

Academic Editor

PLOS ONE

Additional Editor Comments (optional):

The manuscript has improved its quality after revision

Reviewers' comments:

Reviewer's Responses to Questions

**Comments to the Author**

1. If the authors have adequately addressed your comments raised in a previous round of review and you feel that this manuscript is now acceptable for publication, you may indicate that here to bypass the “Comments to the Author” section, enter your conflict of interest statement in the “Confidential to Editor” section, and submit your "Accept" recommendation.

Reviewer #1: All comments have been addressed

Reviewer #2: All comments have been addressed

2. Is the manuscript technically sound, and do the data support the conclusions?

Reviewer #1: Yes

Reviewer #2: (No Response)

3. Has the statistical analysis been performed appropriately and rigorously? 

Reviewer #1: Yes

Reviewer #2: (No Response)

4. Have the authors made all data underlying the findings in their manuscript fully available?

Reviewer #1: Yes

Reviewer #2: (No Response)

5. Is the manuscript presented in an intelligible fashion and written in standard English?

Reviewer #1: Yes

Reviewer #2: (No Response)

6. Review Comments to the Author

Reviewer #1: All questions and comments have been answered. I would like extend my gratitude to the authors for taking the time to respond to the feedback and assimilate them in their work for maximal scientific impact of the paper.

Reviewer #2: (No Response)

7. PLOS authors have the option to publish the peer review history of their article (what does this mean?). If published, this will include your full peer review and any attached files.

Reviewer #1: No

Reviewer #2: No

---

## [Editor Report · Acceptance letter]

1 Jun 2022

PONE-D-21-30166R1 

Dynamic blood oxygen indices in mechanically ventilated COVID-19 patients with acute hypoxic respiratory failure: a cohort study 

Dear Dr. Grocott:

I'm pleased to inform you that your manuscript has been deemed suitable for publication in PLOS ONE. Congratulations! Your manuscript is now with our production department. 

Kind regards, 

on behalf of

Dr. Antonino Salvatore Rubino 

Academic Editor

PLOS ONE